

# PhotoVault
## Aplikacja webowa do prezentowania i sprzedawania fotografii

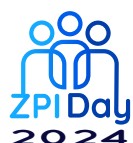

**Autorzy**: Małgorzata Orzeł ⬤ · Michał Trojanowski ⬤

**Opiekun:** Piotr Zabawa

**Streszczenie**

Stworzyliśmy platformę internetową PhotoVault, która umożliwia fotografom prezentację i sprzedaż zdjęć w formie cyfrowej i fizycznej, zapewniając ochronę praw autorskich i rozszerzając możliwości zarobkowe. Platforma oferuje od strony klienta m.in. przeglądanie i filtrowanie zdjęć, dodawanie do koszyka, składanie zamówień, a od strony fotografa: dodawanie i zarządzanie zdjęciami, ustalanie ich ceny i licencji. Platforma jest dostępna w dwóch językach i zapewnia intuicyjną nawigację.

# 1  OPIS PROBLEMU ORAZ CELE PROJEKTU

Fotografowie, którzy chcą monetyzować swoją pracę poprzez sprzedaż internetową, napotykają problem częstego naruszania ich praw autorskich poprzez nielegalne rozpowszechnianie ich zdjęć. Dodatkowo, działając samodzielnie, mają ograniczone możliwości masowej sprzedaży fizycznych kopii swoich prac. Te czynniki sprawiają, że ich potencjalne źródło dochodu jest mocno ograniczone. Mimo tego że istnieją platformy zdjęć stockowych, które uzyskują licencję od autorów i walczą z naruszeniami praw autorskich, to nie rozwiązują one problemu sprzedaży fizycznych wydruków, co dla wielu twórców jest również ważnym aspektem.

Celem projektu jest stworzenie platformy internetowej umożliwiającej fotografom monetyzację ich pracy poprzez sprzedaż zdjęć w formie cyfrowej i fizycznej. Platforma ma zapewnić ochronę praw autorskich, m.in. dzięki zastosowaniu znaków wodnych, oraz rozszerzyć możliwości zarobkowe fotografów, uwzględniając integrację z usługami druku na zamówienie. Projekt ma na celu rozwiązanie problemu nielegalnego rozpowszechniania zdjęć oraz ograniczonych możliwości sprzedaży fizycznych kopii prac.

Zakres przedsięwzięcia

1. Dla fotografów:

   · możliwość dodawania zdjęć, określania licencji oraz ustalania cen sprzedaży zdjęć w formie cyfrowej i fizycznej

   · organizacja zdjęć w kategoriach i etykietach ułatwiających wyszukiwanie (np. gatunek zwierzęcia, efekty świetlne)

   · zgłaszanie nielegalnego umieszczenia zdjęć w serwisie do administracji

2. Dla klientów:

   · wyszukiwanie zdjęć z możliwością filtrowania według kategorii i etykiet

   · zakup zdjęć w formie cyfrowej oraz zamawianie ich w formie fizycznych wydruków

   · możliwość wyboru rozmiaru i materiału wydruku

## 1.1  Prace związane z tematem

W ramach badania rynku zidentyfikowaliśmy praktyki stosowane przez istniejące platformy do sprzedaży zdjęć, aby jak najlepiej dostosować serwis do potrzeb twórców, jak i klientów. W szczególności zwracamy uwagę na możliwość oferowania wydruków fizycznych, ponieważ nasze wstępne analizy wykazały, że ta opcja nie jest tak często spotykana na istniejących platformach, a jednocześnie stanowi istotną wartość dodaną dla użytkowników, którzy cenią sobie możliwość posiadania fizycznej kopii dzieła, jej wyjątkowość i trwałość.

W tabeli 1 prezentujemy analizę wybranych platform internetowych oferujących sprzedaż fotografii online. Przyjrzyjmy się bliżej ich funkcjonalnościom i modelom biznesowym:

- Shutterstock - amerykańskie przedsiębiorstwo [6] zajmujące się sprzedażą zdjęć i filmów stockowych. Klienci mogą nabywać materiały w dwóch modelach: poprzez zakup paczki zdjęć lub subskrypcję. Wykupywany dostęp do zdjęć podlega dwóm kategoriom licencyjnym: licencji standardowej i licencji rozszerzonej. Licencja standardowa nakłada ograniczenia na koszty produkcji w ramach których wykorzystywane są materiały, oraz na nakład (np. drukowanych gazet). Serwis oferuje również zasoby stockowe dostępne bezpłatnie. Model wynagrodzeń fotografów jest oparty na prowizji, której wysokość jest zależna od kilku czynników (np. liczby pobrań, rodzaju licencji) i mieści się ona w granicach 15-40%.

- Unsplash - społecznościowa platforma internetowa [2] przeznaczona do udostępniania fotografii. Wszystkie zdjęcia dostępne w serwisie są darmowe do użytku zarówno w celach osobistych, jak i komercyjnych, bez ograniczeń oraz bez obowiązku wskazywania autora. Zabronione jest natomiast odsprzedawanie zdjęć w ich oryginalnej formie oraz zbieranie zdjęć i umieszczanie ich na innych platformach oferujących zasoby stockowe.

- Fine Art America - internetowa platforma [1] zajmująca się sprzedażą dzieł sztuki (fotografii, obrazów, grafiki komputerowej, itd.). Serwis ten działa na zasadzie rynku online, gdzie klienci wykupują konkretne produkty, a nie subskrypcję, tak jak to ma miejsce na platformach stockowych. Fine Art America oferuje możliwość zamawiania dzieł w różnych formatach oraz na różnorodnych materiałach. Model biznesowy tej platformy opiera się na nakładaniu marży na ceny ustalane przez autorów. Dodatkowo, autorzy mogą wykupić roczną subskrypcję premium, dzięki której twórca otrzymuje dodatkowe narzędzia promocyjne dla swoich dzieł. Obecnie firma nie posiada drukarni w Polsce, co skutkuje wyższymi kosztami wysyłki na nasz rynek.

| Serwis | Koszty klientów | Koszty autorów | Darmowe materiały | Wydruk | Cyfrowe kopie | Zarobki autorów |
|---|---|---|---|---|---|---|
| Shutterstock | Paczki zdjęć lub subskrypcja (miesięczna lub roczna). Licencja standardowa lub rozszerzona. Opcja najtańsza: 2 zdjęcia za €29. Subskrypcja: 10 zdjęć/miesiąc za €29. | Brak | Ograniczona ilość | Nie | Tak | Pomiędzy 15-40% ceny, szacunkowo $0.10 - $5.80 za zdjęcie i $1.25 - $47.92 za wideo |
| Unsplash | Brak | Brak | Tak, wszystkie | Nie | Tak | Brak |
| Fine Art America | Cena ustalona przez autora + marża serwisu | $30 rocznie, subskrypcja premium | Nie | Tak, wybór formatu i materiału. Brak drukarni w Polsce - wysokie koszty wysyłki | Nie | Autorzy sami ustalają cenę |

Tabela 1: Tabela badanie rynku

Analiza wyżej wymienionych platform dostarczających zarówno cyfrowe, jak i fizyczne dzieła wskazuje na różnorodność podejść do modeli biznesowych, licencjonowania zdjęć oraz dostępności produktów. Funkcjonalnościami, które zwróciły naszą największą uwagę, są:

- model finansowy serwisu Fine Art America, w którym fotografowie ustalają własną cenę uznaliśmy za wyjątkowo korzystny dla fotografów, jak i klientów, co stanowi istotny atut w przyciąganiu użytkowników do serwisu.

- dostępność darmowych materiałów oferowana przez serwis Unsplash uważamy za mającą równie duże znaczenie co wyżej wymieniony model, zatem zamierzamy w naszym serwisie umożliwiać fotografom zamieszczanie darmowych zdjęć.

- model subskrypcyjny serwisu Shutterstock jak wynika z naszego doświadczenia, nie jest postrzegany jako atrakcyjny przez klientów, dlatego zdecydowaliśmy się go nie implementować

- możliwość fizycznych wydruków serwisu Fine Art America uważamy za kluczowy element, ponieważ brakuje go aktualnie na polskim rynku i dlatego może się on spotkać z dużym zainteresowaniem użytkowników

Łącząc wybrane funkcjonalności z analizowanych platform, stworzyliśmy unikalną aplikację webową, która odpowiada na zróżnicowane potrzeby zarówno fotografów, jak i klientów.

## 1.2   Wykorzystane technologie i napotkane ograniczenia

Przy tworzeniu projektu posługiwaliśmy się technologiami takimi jak:

- Figma [4] – narzędzie do projektowania interfejsów aplikacji.

- Jira [9]– narzędzie do zarządzania projektami i planowania zadań.

- Next.js [7] – framework oparty na React do budowy aplikacji webowych z dodatkowymi funkcjami wspomagającymi SEO.

- TypeScript [5]– język programowania używany do tworzenia aplikacji.

- Prisma [8] - framework ORM do Next.js.

- GitHub [3] - współdzielenie kodów źródłowych i artefaktów.

Realizacja projektu napotkała istotne ograniczenia, przede wszystkim w zakresie dostępnego czasu oraz zasobów ludzkich. Dwuosobowy zespół projektowy, pomimo wysokich kompetencji i zaangażowania, musiał zmierzyć się z napiętym harmonogramem, co wpłynęło na ograniczenie zakresu projektu i rezygnację z pewnych funkcjonalności, m.in. ograniczenie kontaktu klienta z fotografem do korespondencji mailowej zamiast chatu dostępnego na stronie, forum, na którym fotografowie lub organizatorzy wystaw mogliby zamieszczać dostępne oferty. Niemniej jednak, dzięki efektywnej organizacji pracy i wybraniu najważniejszych funkcjonalności do realizacji, udało się stworzyć w pełni działającą aplikację pokrywającą cele wyznaczone na samym początku.

## 1.3   Wyniki

Nasz zespół z powodzeniem zaimplementował szereg kluczowych funkcjonalności:

- przeglądanie oraz filtrowanie zdjęć po kategorii, cenie oraz etykietach (tagach)

- dodawanie wybranych zdjęć do koszyka aby móc złożyć zamówienie na więcej niż jedno zdjęcie na raz

- możliwość rejestracji i logowania jako klient oraz fotografa z danymi logowania przesyłanymi za pomocą tokenu JWT dla poprawienia bezpieczeństwa

- strony profilowe fotografów z opisem o nich, opcjonalnym adresem email do kontaktów z klientem oraz ich galerią zdjęć

- dla fotografów możliwość dodawania oraz edycji własnych zdjęć z etykietami, ustalania ich cen, dostępnych licencji oraz ich widoczności w serwisie

- zgłaszanie nielegalnie zamieszczonych zdjęć przez fotografów

- promowanie zdjęć i fotografów na stronie głównej

- intuicyjny interfejs stworzony w kontrastowych odcieniach czerni i bieli oraz w dwóch językach: polskim i angielskim pozwala na łatwą nawigację i dostęp do wszystkich funkcji serwisu dla większego grona klientów

## 2   WNIOSKI

W ramach projektu stworzyliśmy platformę internetową ułatwiającą kontakt pomiędzy fotografami a klientami zainteresowanymi zakupem ich zdjęć.

Zależało nam na tym, aby serwis był dostępny dla szerokiego grona odbiorców, dlatego stworzyliśmy go:

- jako aplikacje webową, co daje do niej dostęp na dużej ilości urządzeń elektronicznych,

- użycia na niej oprócz języka polskiego także angielskiego, który jest szeroko używany,

- oraz przejrzystego i kontrastowego interfejsu, który zwiększa dostępność internetową.

Mamy nadzieję, że stworzona platforma przyczyni się do rozwoju rynku fotograficznego w Polsce, ułatwiając artystom dotarcie do szerszej publiczności i monetyzację ich pracy. Jesteśmy przekonani, że serwis spełni oczekiwania zarówno fotografów, jak i klientów, oferując im wygodne i bezpieczne narzędzie do prezentacji, zakupu i sprzedaży zdjęć.

## 2.1 Kierunki rozwoju

Stworzona platforma stanowi solidny fundament do dalszego rozwoju i rozbudowy, szczególnie związany z dynamicznie zmieniającym się rynkiem.

W przyszłości chcielibyśmy rozszerzyć ją o wybrane funkcjonalności:

- integrację z mediami społecznościowymi

- narzędzia do analizy sprzedaży, system rekomendacji na podstawie dokonanych wcześniej już zakupów

- forum, na którym fotografowie lub organizatorzy wystaw mogliby zamieszczać dostępne oferty

- program lojalnościowy dla stałych klientów.

Wprowadzenie tego przyciągnęłoby większą liczbę użytkowników, zwiększyłoby zaangażowanie istniejących klientów i umożliwiło artystom dotarcie do szerszej publiczności. W dalszej perspektywie platforma mogłaby stać się wiodącym miejscem spotkań fotografów i miłośników fotografii, promując twórczość i ułatwiając jej dostępność.

## LITERATURA

[1] Sean Broihier. Fine art america. https://fineartamerica.com/. [Data uzyskania dostępu: 9 listopada 2024].

[2] Mikael Cho. Unsplash. https://unsplash.com/. [Data uzyskania dostępu: 9 listopada 2024].

[3] Tom Preston-Werner P. J. Hyett Chris Wanstrath, Scott Chacon. Github. https://www.github.com/. [Data uzyskania dostępu: 18 października 2024].

[4] Dylan Field. Figma. https://www.figma.com/. [Data uzyskania dostępu: 18 października 2024].

[5] Anders Hejlsberg. Typescript. https://www.typescriptlang.org/. [Data uzyskania dostępu: 18 października 2024].

[6] Jon Oringer. Shutterrstock. https://www.shutterstock.com/. [Data uzyskania dostępu: 9 listopada 2024].

[7] Guillermo Rauch. Next.js. https://www.nextjs.org/. [Data uzyskania dostępu: 18 października 2024].

[8] Søren Bramer Schmidt. Prisna. https://www.prisma.io/. [Data uzyskania dostępu: 18 października 2024].

[9] Mike Cannon-Brookes Scott Farquhar. Jira. https://www.atlassian.com/software/jira. [Data uzyskania dostępu: 18 października 2024].
