# OpenReview forum: "PhotoVault"
_pwr.edu.pl/Wrocław_University_of_Science_and_Technology/2024/ZPI_Day — Wrocław University of Science and Technology 2024 ZPI Day Submission_

### Official Review · Reviewer_CpMi · 2024-12-03
**Zbyt ogólnikowy opis opracowanego rozwiązania**

**Confidence:** 5
**Significance Of Results:** 3
**Overall Quality:** 3

**Compliance With Template:**

3: Average Quality – The article includes most of the required sections, but some may be incomplete, written in a general or unclear manner. The content is correct but requires further refinement.

**Description Of Results:**

2: Low Quality – The results are described very superficially and in a general manner. Essential details, usage examples, or evaluations are missing.

**Feedback On Consistency:**

- W zdaniu "Celem projektu jest stworzenie platformy internetowej" użyto czas przyszły? Natomiast dalej autorzy posługują się czasem przeszłym (rozdział 1.2, 1.3). Artykuł jest opisem wykonanych prac, zatem czas przyszły jest tu nie wskazany.

- zbyt obszerny opis poświęcony systemom konkurencyjnym, wystarczyłaby tabela zestawiająca funkcjonalności konkurencji względem rozwiązania Studentów.

- na czym polega doświadczenie Studentów wyrażone w zdaniu: "model subskrypcyjny serwisu Shutterstock jak wynika z naszego doświadczenia, nie jest postrzegany jako atrakcyjny przez klientów"?

- w zdaniach: "Łącząc wybrane funkcjonalności z analizowanych platform, stworzyliśmy...", "...udało się stworzyć...", "W ramach projektu stworzyliśmy ..." autorzy przypisują sobie Boską władzę stwarzania. W tym kontekście słowo to jest błędne, bo stwarzanie odbywa się z niczego. Zatem poprawnie byłoby napisać "opracowaliśmy", "wykonaliśmy" lub "zaprojektowaliśmy".

- Zdanie/podpunkt w rozdziale 2: "Zależało nam na tym, aby serwis był dostępny dla szerokiego grona odbiorców, dlatego stworzyliśmy go: -> użycia na niej oprócz języka polskiego także angielskiego, który jest szeroko używany" jest niepoprawnie językowo sformułowane. Brzmi źle, można było napisać: "zaimplementowano wielojęzykowy interfejs użytkownika". Niestety nie jest to jedyne miejsce tego typu.

- W zdaniu: "Jesteśmy przekonani, że serwis spełni oczekiwania zarówno fotografów, jak i klientów, oferując im wygodne i bezpieczne narzędzie do prezentacji, zakupu i sprzedaży zdjęć." autorzy wspominają o bezpiecznym narzędziu - bez rozwinięcia na czym owo bezpieczeństwo ma polegać. To samo tyczy się słowa "wygodne" - nie wiadomo z artykułu na czym ta wygoda ma polegać.

**Potential For Development:**

Choć aktualny stan zaawansowania projektu wskazuje, że rozwiązanie nie jest w stanie podjąć realnej walki z konkurencyjnymi systemami, to na pewno projekt posiada potencjał, aby zostać rozwinięty do dojrzałego rozwiązania.

**Project Nature Evaluation:**

- Opis wyników jest minimalny. Posłużę się stwierdzeniem, które względem tytułu projektu ma podwójne znaczenie: "Obraz jest wart tysiąca słów". W artykule nie przedstawiono choćby jednego zdjęcia opracowanej aplikacji. Celem projektu było zaprojektowanie i wykonanie systemu webowego, zatem aż prosi się, aby załączyć kilka zrzutów ekranu przedstawiających najważniejsze funkcjonalności aplikacji.

- Trudno jest ocenić wagę znaczenia tego projektu z uwagi na minimalny opis sposobu jego wykonania. Nie sposób ocenić, czy proponowane rozwiązanie jest rzeczywiście konkurencyjne względem istniejących serwisów. Z opisu nie wynika jednoznacznie przewaga nad konkurencją.

- W opisie projektu całkowicie pominięto rodzaj architektury zastosowany do implementacji. Wspomniano jedynie lakonicznie, że wykorzystano Next.js. Do których części aplikacji (Frontend/Backend)?

- Brak jest także opisu najistotniejszych aspektów projektów tego rodzaju. Nie przedstawiono informacji w jaki sposób są ładowane zdjęcia na serwer (pojedynczo, wiele plików na raz, wsadowo, etc.). Nie wspomniano o sposobie przechowywania zdjęć i ich przetwarzania. Nic nie wiadomo, czy generowane są miniaturki widoczne na stronie przed zakupem. Nie opisano jak system zabezpiecza zdjęcia przed ich wykradaniem. Nie wiadomo jak zdjęcia są dostarczane po zakupie do klienta. Proces zakupu także jest nie opisany.

- Autorzy sami wspominają, że pewne funkcjonalności nie udało się zaimplementować w zadanym czasie. Takie samo oskarżanie się w artykule jest nie wskazane. Po co odbierać pracę recenzentom ;).

- Brak jest także opisu kluczowej funkcjonalności, która miała odróżniać projekt od istniejących serwisów. Na czym polega wydruk zdjęć? Czy chodzi o odbitki foto, wydruki wielkopowierzchniowe, nadruki na kubkach ?

**Technical Language Precision:**

2: Low Quality – The language is partially inappropriate. Significant terminology errors and numerous ambiguities are present. Some sections are imprecise or inconsistent with the expected style of a technical report.

---

### Official Review · Reviewer_8o2U · 2024-12-04
**PhotoVault - Aplikacja webowa do prezentowania i sprzedawania fotografii**

**Confidence:** 5
**Significance Of Results:** 5
**Overall Quality:** 5

**Compliance With Template:**

5: Very High Quality – The article contains all the required sections, which are written in a very detailed, clear, and error-free manner. The structure is professional and meets expectations, and the content adheres to the highest substantive and formal standards.

**Description Of Results:**

4: High Quality – The results are described in detail and supported by usage examples or evaluations. The description is reliable but may lack full depth of analysis.

**Feedback On Consistency:**

The paper is written very well but one lacking element is a presentation of the approach to evaluation and verification of the software system, like at least a test strategy or a concept of validation. However, the problem may result from the fact that the team started their project two weeks later and has limited number of members. Moreover, the paper was written before the tests were implemented and executed - perhaps to early to be completed in the mentioned aspect.

**Potential For Development:**

Definitely yes, there is real market opportunity, which was identified correctly. The choice of functionalities is limited and their choice is very good for the first release of the product. There is also a chance for evolution of the product in the future. The aspect of potential development of the  product is widely described in the document.

**Project Nature Evaluation:**

The approach to the project is very professional – the subject was motivated and positioned very well on the market, the market opportunity for the solution was identifed correctly. Almost all technical aspects of the project (except the test discipline related) were implemented correctly and their presentation has high quality.

**Technical Language Precision:**

5: Very High Quality – The language is entirely appropriate for a technical report. All terms are used correctly and precisely, and the style is professional, clear, and coherent, without any errors or ambiguities.

---

### Official Review · Reviewer_ZJF6 · 2024-12-04
**Rekomendowany do publikacji z drobnymi poprawkami**

**Confidence:** 5
**Significance Of Results:** 4
**Overall Quality:** 4

**Compliance With Template:**

5: Very High Quality – The article contains all the required sections, which are written in a very detailed, clear, and error-free manner. The structure is professional and meets expectations, and the content adheres to the highest substantive and formal standards.

**Description Of Results:**

4: High Quality – The results are described in detail and supported by usage examples or evaluations. The description is reliable but may lack full depth of analysis.

**Feedback On Consistency:**

Tekst jest spójny i logiczny, choć niektóre przejścia między sekcjami mogłyby być płynniejsze.

Uwago:
1. "Zakres przedsięwzięcia" - Proponuję zrobić z tego akapit lub zintegrować to z listą poniżej.
2. Listy, wszystkie - proponuję użyć przecinków na końcu poszczególnych punktów i kropki w ostatnim punkcie.
3. Tabela 1: Tabela badanie rynku - Teksty w tabeli nachodzą na ramkę. Kolumny nie muszą mieć takiej samej szerokości - wtedy uda się ładniej ułożyć w nich zawartość.
4. "Realizacja projektu napotkała ", "Dwuosobowy zespół projektowy... musiał" - ta trzecia osoba brzmi trochę sztucznie. Ten zespół to wy. To wasze problemy i wyzwania. Nie ma co przesadzać z bezosobowością.
5. Artykuł kończy się w połowie strony. Może warto by było wykorzystać to miejsce na jeden/kilka screenshotów z aplikacji? Po ułożeniu tabelki będzie jeszcze więcej miejsca.

**Potential For Development:**

Bardzo dobrze opisane i przemyślane kierunki rozwoju.

**Project Nature Evaluation:**

Projekt ma znaczenie praktyczne i, być może, potencjał biznesowy. Rozwiązanie w świetle przedstawionej konkurencji wydaje się być innowacyjne. Dobrze opisano ograniczenia i możliwości wdrożenia. Użyto adekwatnych narzędzi i frameworków.

"możliwość rejestracji i logowania jako klient oraz fotografa z danymi logowania przesyłanymi za pomocą tokenu JWT dla poprawienia bezpieczeństwa" - Faktycznie w JWT mieliście "dane logowania"? Technicznie jest to możliwe. Tylko pytanie po co i czy faktycznie były to "dane logowania"? Warto to wyjaśnić.

**Technical Language Precision:**

4: High Quality – The language is appropriate for a technical report. Terminology is used correctly, and statements are precise, with only minor shortcomings that do not affect the overall clarity.

---

### Decision · Program_Chairs · 2024-12-10

Accept (Poster)